**Data Availability Statement:** All relevant data are within the manuscript and its Supporting

# Healthcare provider and patient/family perceptions of continuous pressure imaging technology for prevention of pressure injuries: A secondary analysis of patients enrolled in a randomized control trial

Wrechelle Ocampo[1]*, Darlene Y. Sola[1], Barry W. Baylis[1,2,3], John M. Conly[1,2,3,4], David B. Hogan[1,2], Jaime Kaufman[1], Linet Kiplagat[1], Henry T. Stelfox[1,2,3,5], William A. Ghali[1,2,3], Chester Ho[1,2,3,6]

1 W21C, O'Brien Institute for Public Health, University of Calgary, Calgary, AB, Canada, 2 Department of Medicine, Cumming School of Medicine, University of Calgary, Calgary, AB, Canada, 3 Alberta Health Services, Calgary, AB, Canada, 4 Synder Institute for Chronic Diseases, Cumming School of Medicine, Calgary, AB, Canada, 5 Departments of Critical Care Medicine and Community Health Sciences, Calgary, AB, Canada, 6 Division of Physical Medicine & Rehabilitation, Department of Medicine, Faculty of Medicine & Dentistry, University of Alberta, Edmonton, AB, Canada

* wbocampo@ucalgary.ca

## Abstract

### Introduction

Despite the availability of various pressure injury (PI) prevention strategies (e.g., risk identification, use of pressure re-distribution surfaces, frequent repositioning), they persist as a significant issue for healthcare systems worldwide. Continuous pressure imaging (CPI) is a novel technology that could be integrated within a comprehensive approach to the prevention of PIs. We studied the perceptions of healthcare providers and patients/families to identify facilitators and barriers to the use of this technology.

### Methods

Hospitalized patients/family members from a randomized controlled trial assessing the efficacy of CPI in preventing PIs completed a survey after 72 hours (or upon discharge from hospital) of CPI monitoring. They were asked questions about prior and current experience with CPI technology. For healthcare providers, perceptions on the use of the device and its impact on care were explored through a survey distributed by email or hard copies.

### Results

A total of 125 healthcare providers and 525 patients/family members completed the surveys. Of the healthcare providers, 95% either agreed/strongly agreed that the CPI technology was easy to use and 65% stated that the device improved how they provided pressure relief for patients. Identified issues with the device were cost, the fitting of the mattress cover, and the fixation of the patients/families on the device. Over a quarter of the patient/family

Information files. Data are available from the University of Calgary Institutional Data Access (contact via w21c@ucalgary.ca) for researchers who meet the criteria for access to confidential data. Data is made available upon request due to legal restrictions based on University of Calgary policies sharing research data with external researchers.

**Funding:** The study is funded by Alberta Innovates Health Solutions' Collaborative Research and Innovation Opportunities grant. The grant has been awarded to WG, JC, and TS under grant number 201201137. The funders were only involved in approval of the study design. They were not involved in development of the study design, data collection, analysis, decision to publish, or preparation of the manuscript. Information on the funder can be found at https://albertainnovates.ca/what-we-offer/funding-grants/.

**Competing interests:** The authors have declared that no competing interests exist.

respondents agreed/strongly agreed that the device influenced how pressure relief was provided. This response was statistically associated with whether the monitor was turned on (intervention arm; 52.7%) or off (control arm; 4.2%).

## Discussion and conclusion

CPI technology was positively perceived by healthcare providers. Most patients/families felt it influenced care when the CPI monitor was turned on. Concerns raised around cost and the ease of use of these devices by healthcare providers may affect the decisions of healthcare system administrators to adopt and implement this technology.

## Introduction

Standard practices for preventing pressure injuries include risk assessment, nutritional supplementation, frequent repositioning and use of pressure re-distribution surfaces [1, 2]. However, pressure injuries continue to affect one out of ten adults who are hospitalized globally [3] and are associated with significant costs to healthcare systems [3, 4].

Technologies such as continuous pressure imaging (CPI) could be integrated into a comprehensive approach to the prevention of pressure injuries and if effective, prove beneficial for healthcare systems. CPI helps healthcare providers identify areas of high and/or prolonged interface pressure. This information can then be used to better inform the repositioning of patients to provide pressure relief and help prevent the development of pressure injuries [5, 6]. However, this technology is relatively costly [3], which might influence its widespread adoption.

The perceptions of healthcare providers, patients, and family members are important in understanding the acceptability of CPI technology [7–10]. Perceptions captured about other continuous monitoring technologies in healthcare have helped to identify barriers to their implementation by healthcare staff. These barriers can include reservations about the utility of the technology and doubts about their perceived value in patient care [7–10]. We were only able to identify one study on the perceptions of healthcare providers about CPI technology [11] and none that captured patient or family perspectives. For these reasons, we felt there was a need to further examine user perceptions about CPI and its utility.

Capturing the perceptions of both hospitalized patients or their family members and healthcare providers about CPI was a secondary objective of a randomized controlled trial (RCT) on the efficacy of this technology in preventing pressure injuries that we recently conducted (for additional data about the trial please see Wong et al [12]). In this paper, we report on the opinions of healthcare providers and patients or family members about CPI.

## Methods

Human Subject Research (involving human participants).

CHREB (Conjoint Health Research Ethics Board) approved the study. The research ID# is REB13-0794. Approval is granted only for the project and purposes.

Written consent was obtained.

### Study setting

The main study was an RCT that assessed the effect of the ForeSite PT™ system (XSESNSOR Technology Corp., Calgary, AB, Canada) on reducing interface pressure and soft tissue changes that could lead to pressure injury [12]. The system consisted of two parts: a flexible,

thin mattress cover with embedded sensors that was positioned under a fitted hospital sheet; and, a liquid crystal display (LCD) monitor that was mounted at the head of the bed and displayed the information the sensors received. The LCD monitor displayed in colour and real-time pressure areas as the patient lay on their bed. Included in the pressure data on the monitor was a clock that had a default setting of two hours and would count down to indicate when the next turn or reposition was due. There was no audible alarm for this feature–there was only a notification on the LCD display. The clock could be set for more frequent turn notification if the healthcare worker wished to adjust the setting. Nursing staff were educated on the device prior to study recruitment during short education sessions. They were also emailed a link to an instructional video and each unit was provided with detailed instructions on the device. Systems had software updates by the developer prior to initiation of study recruitment and half-way through study recruitment.

The trial occurred in a tertiary acute care hospital. Nursing units which had patients at higher risk of pressure injury formation were the sites of patient recruitment. These included internal medicine, neurology, neurosurgery, nephrology, intensive care and cardiovascular intensive care units.

## Sampling and subject recruitment

For the RCT, the calculated required sample size was based on the assumption of a 33% relative risk reduction in pressure injuries, an alpha level of 0.05 and 80% power. This led to an anticipated need for 678 randomized patients with an expected attrition proportion of 12% for the primary study outcome [12]. In a secondary exploratory analysis, a convenience sample made up of eligible hospitalized patients (or family members) and healthcare providers was used to evaluate perceptions about CPI.

Eligible inpatient adults and their families/caregivers who consented to the RCT [12], were asked about their perceptions of the device after 72 hours or on the day of their hospital discharge. Healthcare workers who cared for participants were also surveyed on the products' usability, their interaction with the system, and the sense of the effectiveness of the ForeSite PT™ in reducing pressure injuries.

The intervention group had the ForeSite PT™ system set up with the monitor on and displayed for use by healthcare workers. The control group had the ForeSite PT™ system turned on for collecting continuous interface pressure data but the LCD monitor was covered, and the settings were adjusted to low sensitivity, such that the visual feedback to the healthcare providers even, if the monitor was turned on, would not highlight areas of the body in need of pressure relief. For control patients, the healthcare worker provided standard of care to the patient without visual feedback on pressure recordings. During the process of informed consent, the benefit of the intervention over standard of care and randomization to either group was explained to the participant or surrogate.

## Content of surveys

Researchers preferentially surveyed the patient but if they were unable to answer, family members/caregivers were approached about their perceptions of the device. They were asked one open-ended and six multiple choice questions about prior and current experience with CPI technology. These questions included asking about the care provided and how comfortable the device was to use. The questions took less than 10 minutes to complete. Survey responses were entered and kept in an electronic database [13].

Healthcare providers were asked separately about their experiences using the CPI system. Their survey included 12 multiple choice and five open-ended questions about the

functionality and ease of both use and interpretation of the pressure data provided on the LCD screen. They were questioned about any past experience with this kind of technology. The survey was administered approximately three months after the staff were exposed to the CPI device and had cared for several patients using the equipment (i.e., all cared for one or more patients in the intervention arm). Healthcare providers were invited by email to complete the survey online [13] or asked in-person by the research team to complete a hard copy of the survey instrument.

There was no formal validation process for the surveys. Both surveys were developed, reviewed, and piloted within the study team. Contact information of the study team was provided in the email invitation to the survey in case healthcare providers had questions, or the study team was available to answer questions when the survey was distributed in-person. There were no concerns or questions about the surveys by either groups, and respondents were not obligated to answer questions that they were uncomfortable answering. Copies of both surveys are provided in (S1 and S2 Files).

## Analysis

Multiple choice questions were analyzed using descriptive statistics performed on Microsoft Excel, Version 2020. Where deemed appropriate chi-square testing was done. Open-ended questions and any free text responses of closed-ended questions were analyzed using inductive thematic analysis [14]. These responses were coded manually in Microsoft Excel, Version 2020 and then grouped into themes based on similarities of responses.

## Results

### Healthcare provider survey

The healthcare provider survey was distributed to over 350 healthcare providers with 125 (35%) responses obtained (the majority were from nurses) (Table 1). For some questions, not

**Table 1. Characteristics of healthcare providers who completed the survey.**

| Question and responses | N | % |
|---|---|---|
| **Please specify your profession** | | |
| [1]Clinician | 2 | 1.6 |
| Healthcare aid | 6 | 4.8 |
| Nurse | 114 | 91.2 |
| Other | 1 | 0.8 |
| Not indicated | 2 | 1.6 |
| **Have you used any pressure-sensing mattress system prior to this study?** | | |
| Yes | 24 | 19.2 |
| No | 98 | 78.4 |
| Not indicated | 3 | 2.4 |
| **On what type of unit did you use the pressure-sensing mattress system for this study?** | | |
| Internal medicine | 14 | 11.2 |
| Nephrology | 16 | 12.8 |
| Stroke | 10 | 8.0 |
| Spine | 10 | 8.0 |
| Neurology | 8 | 6.4 |
| Intensive care | 13 | 10.4 |
| Not indicated | 54 | 43.2 |

[1]Nurse practitioners, residents and attending physicians

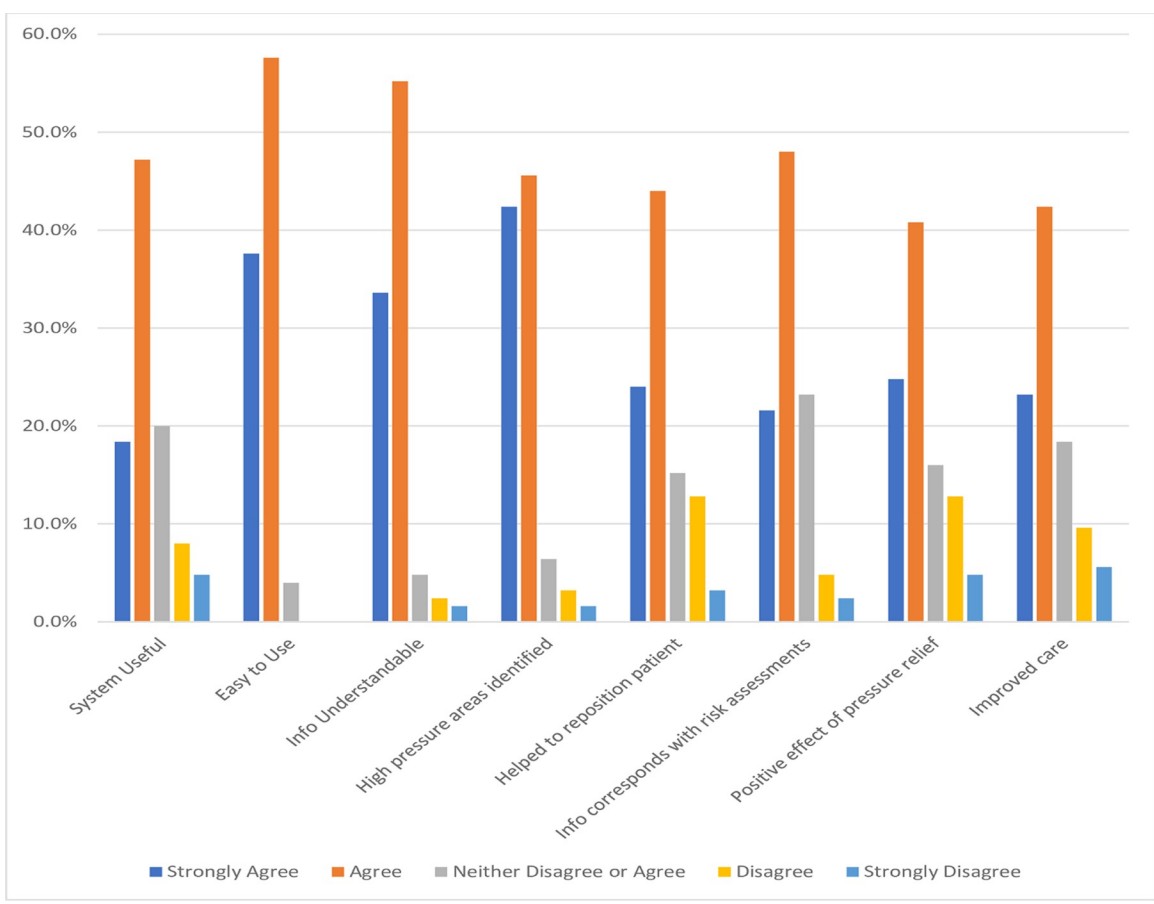

**Fig 1. Health care providers responses to multiple choice survey questions.**

all healthcare providers provided answers, which accounts for why not all percentages in Table 1 total 100%.

Fig 1 shows the responses to the multiple choice questions about the pressure-sensing mattress cover system with a Likert scale used for responses. Approximately half of healthcare providers agreed that the system was useful, easy to use, and the information provided by it was easy to understand. Almost half of the healthcare providers agreed that with its use, they were able to identify high areas of pressure and this information helped in repositioning the patient appropriately. It was felt the information provided by the device corresponded with other risk assessments for developing pressure injuries they performed (e.g., Braden scale) [15]. For instance, areas in need of pressure relief might also need more attention for moisture control and reducing friction/shear. Almost half of the healthcare providers also agreed that the device positively affected the way pressure relief was provided and improved patient care. Features favoured by more than half of respondents were the reset button, timer countdown and patient display area (Fig 2).

Table 2 summarizes the open-ended question responses obtained from the healthcare provider survey.

Noted advantages included improving the repositioning of the patient, a reminder to turn the patient, allowing the patient to sleep more at night, and using the device to help coordinate tasks between staff. While there were no negative impacts on patient care provided, some respondents felt that the device did not change the care they provided.

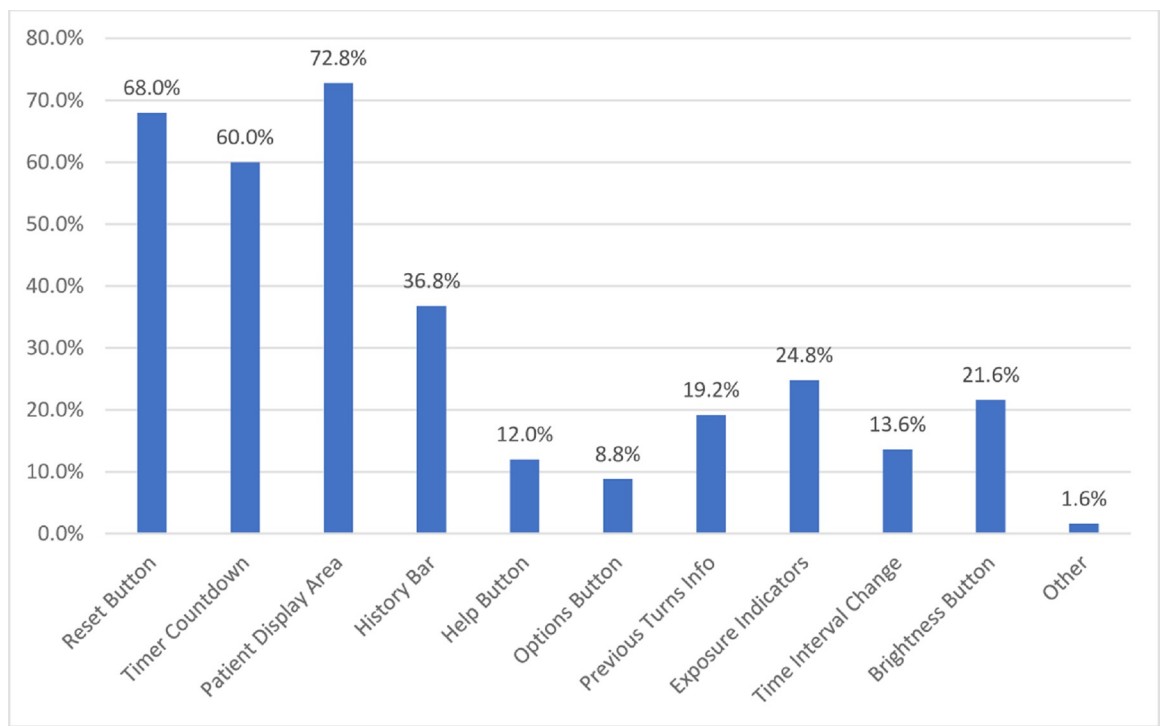

**Fig 2. Features that healthcare providers liked about the pressure-sensing mattress cover system.**

When healthcare providers were asked what they disliked about the device, they noted certain design features, such as the wrinkling and fitting of the mattress cover. They also commented on how they felt it increased their workload, possibly led to reliance on technology and they did not like how patients and families became fixated on the device. System disadvantages noted, included the cost of the device.

Healthcare providers were also asked about ways to improve the device (Table 2). They felt the monitor could be smaller, the device could be made wireless, proposed changes to the location of the monitor, transmission of the information collected to nursing stations, and better identification of areas where repositioning was not effective in relieving pressure. Additional comments obtained included appreciating how the study team set up the device, promotion of staff accountability to turn the patient, and further emphasis on keeping track of turns, and utility for immobile patients.

## Patient/Family survey

Out of 678 patients recruited, a total of 525 respondents completed this survey. There were 154 patients/family members/caregivers who were unable to complete the survey either due to the death of the patient or being discharged from the unit before the survey could be completed. Not all questions were answered, as shown in Table 3.

Most respondents had not used the device before. Notwithstanding this few requested that the sensor mat be removed (n = 14, 2.7%) or that the LCD monitor be turned off (n– 2, 0.4%). When asked how comfortable the mattress cover was to lie or sleep on, 51.2% agreed that it was comfortable while 43.9% neither agreed nor disagreed.

Table 4 compared the Control and Intervention group responses to the statement, "the pressure relief provided by my nurse was influenced by the system". When the monitor was on

**Table 2. Responses to open-ended questions of healthcare provider survey.**

| Question | Themes | Quotes |
|---|---|---|
| **What features did you not like about the system** | Design and hardware | *"Pad wrinkles or bunches"* |
| | | *"Monitor big and in the way"* |
| | | *"Pad didn't always fit mattress"* |
| | | *"Placement at head of bed"* |
| | | *"Fitted sheet doesn't always fit on pad"* |
| | Behavior of healthcare workers | *"Didn't change care"* |
| | | *"Reliance on technology"* |
| | | *"Forget to hit reset button"* |
| | | *"Increases workload"* |
| | Behavior of family/ patient | *"Patient/family fixated on timer"* |
| **What are the advantages of using the pressure-sensing mattress system?** | Outcome of patient | *"Prevention of bed sores"* |
| | | *"More sleep for patients at night"* |
| | | *"Reassured family of patient care"* |
| | Behavior of healthcare workers | *"Remind nurses to turn"* |
| | | *"Better repositioning"* |
| | | *"Helped coordination among staff"* |
| | | *"Improve awareness"* |
| | | *"Adheres to best practice"* |
| | | *"Used for teaching patients"* |
| **What are the disadvantages of using the pressure-sensing mattress system** | Cost | *"Costly"* |
| | Inconvenience | *"Inconvenient to set up"* |
| | | *"more time spent to reposition patient appropriately"* |
| | | *"Monitor in the way"* |
| | | *"Another equipment"* |
| | Affect on patient/ family | *"Pad uncomfortable for patients"* |
| | | *"Patient/family fixated on monitor"* |
| | | *"Unable to alleviate pressure no matter what"* |
| **Please explain any differences on the impact on patient care from the use of the pressure-sensing mattress system compared to not using the system** | Turning patient | *"Visualizing pressure and relief"* |
| | | *"More timely turn"* |
| | | *"Visual reminder"* |
| | | *"Turning patient more frequently"* |
| | Indifferent | *"Doesn't change patient care"* |
| | Awareness | *"more accountable for patient's care"* |
| | | *"pts/family more involved in care"* |
| **How would you improve the pressure-sensing mattress system (e.g. design, information to include, location of monitor, use of 'patient turn' button, etc.)?** | Hardware | *"make monitor a little bit smaller"* |
| | | *"thinner mattress cover"* |
| | | *"make the system be battery operated/wireless"* |
| | Design | *"change the location of the monitor"* |
| | Software | *"download information from monitors directly to nursing station computers"* |
| | | *"highlight pressure points that don't change with turns"* |
| **Additional comments** | Support from study team | *"great for study team to set-up system"* |
| | Accountability | *"staff more accountable"* |
| | Patient care | *"keeping track of patient turns"* |
| | | *"should be used for all immobile patients"* |

**Table 3. Responses to multiple choice questions of patient/family/caregiver survey.**

| Question and responses | N | % |
|---|---|---|
| **Person filling out this form** | | |
| Patient | 305 | 44.9% |
| Family | 114 | 16.8% |
| Community Caregiver | 6 | 0.9% |
| Patient and Family/Caregiver | 10 | 1.5% |
| Unable to complete | 154 | 22.6% |
| Proxy not specified | 80 | 11.8% |
| Question not completed | 10 | 1.5% |
| **Had used similar system before** | | |
| Yes | 2 | 0.4% |
| No | 506 | 96.2% |
| Question not completed | 17 | 3.2% |
| **The sensor mat was comfortable to lie/sleep on** | | |
| Strongly Agree | 54 | 10.3% |
| Agree | 215 | 41.0% |
| Neither agree nor disagree | 231 | 44.0% |
| Disagree | 20 | 3.8% |
| Strongly Disagree | 4 | 0.8% |
| Question not completed | 1 | 0.2% |
| **The sensor mat moved significantly when I laid/slept on it** | | |
| Strongly Agree | 1 | 0.2% |
| Agree | 17 | 3.2% |
| Neither agree nor disagree | 120 | 22.9% |
| Disagree | 293 | 55.8% |
| Strongly disagree | 90 | 17.1% |
| Question not completed | 4 | 0.8% |
| **The pressure relief provided by my nurse was influenced by the system** | | |
| Strongly Agree | 35 | 6.7% |
| Agree | 113 | 21.5% |
| Neither agree nor disagree | 294 | 56.0% |
| Disagree | 76 | 14.5% |
| Strongly disagree | 3 | 0.6% |
| Question not completed | 4 | 0.8% |
| **Did you request that the sensor mat be removed?** | | |
| Yes | 14 | 2.7% |
| No | 511 | 97.1% |
| Question not completed | 0 | 0.0% |
| **Did you request that the LCD monitor be turned off?** | | |
| Yes | 2 | 0.4% |
| No | 520 | 99.0% |
| Question not completed | 3 | 0.6% |

most respondents (137/260, 52.7%) agreed/ strongly agreed that the system influenced pressure injury care compared to less than one in twenty (11/261, 4.2%) when the monitor was off (chi-square statistic 157, $p < 0.001$).

**Table 4. Distribution of responses between the control and intervention groups to the statement, "the pressure relief provided by my nurse was influenced by the system".**

| The pressure relief provided by my nurse was influenced by the system | Control (Monitor OFF) | Intervention (Monitor ON) | Sub-Total |
|---|---|---|---|
| Strongly Agree | 1 | 34 | 35 |
| Agree | 10 | 103 | 113 |
| Neither Agree nor Disagree | 187 | 107 | 294 |
| Disagree | 60 | 16 | 76 |
| Strongly Disagree | 3 | 0 | 3 |
| **Sub-Total** | 261 | 260 | **521** |

## Discussion

One of our research questions in the RCT on the efficacy of CPI was to understand the perceptions of healthcare providers, patients, and their families about this technology. Overall, health care providers reported positive perceptions regarding the device and patients found the technology acceptable.

Nursing staff were the primary healthcare users of the device. Most of them indicated that the device was easy to use and improved how they provided pressure relief for their patients. Many of the nursing staff used the visual feedback from the monitor to target areas of the body that needed pressure relief. Use of the visual feedback was also a response indicated by staff interviewed in a qualitative study by Gunninberg et al [11]. Responses from our survey also indicated nursing staff liked the reminder to turn the patient, how it increased awareness of potential pressure injuries and felt it helped them to coordinate their tasks with other staff. Accountability and reassurance on the provision of patient care directed at preventing pressure injuries were additional advantages mentioned by healthcare providers. These are common perceptions about continuous monitoring devices other than CPI [7, 8].

Some healthcare providers, though, were indifferent or had negative perceptions about the CPI device. These individuals felt that they provided the same quality of patient care with or without the device by routinely assessing and repositioning patients. There were also some disadvantages noted. Similar to Gunninberg et al [11], there were issues with the device around the fitting and creasing of the mattress cover. Some healthcare providers disliked how the patient or family member became fixated on the device when high areas of pressure appeared. They wanted the patient to be turned immediately. More patient and family education about the device and how it is used to inform but not dictate care may alleviate this concern. The cost and the inconvenience of setting up the device were other disadvantages noted that have to be addressed when considering possible implementation of the device.

Perceptions of patients and their families on the CPI technology are also important to consider. Our findings indicate that it was not bothersome and corresponds to the comments made by some that they did not notice the mattress cover was on the bed. When the monitor was turned on, most patients and families agreed or strongly agreed that it influenced pressure injury care, while very few felt it did when the monitor was turned off. This in turn corresponds to the additional comments made that it depended on the nurse to use the device.

There were limitations to our study. Unlike interviews, surveys do not allow the researcher to explore the responses provided to more fully understand them. What lays behind some of the responses received may be mis-interpreted considering this limitation. Nearly a fifth of healthcare providers had used the CPI technology before. This prior use of the device may have influenced their responses on the use of the device in this study with an unconscious bias. Another limitation is the possibility of desirability bias since the study team was asking the participants the survey questions; however, asking the questions, as opposed to the participant

self-completing the survey, allowed for a high response rate and simple yes/no answer questions were avoided. Desirability bias was limited for the healthcare provider surveys by maintaining anonymity. Lastly, while some patient and family education on the technology was provided, it may have been insufficient to allay their concerns and indicate how the information obtained would be used by the nurses providing care. This education may have led to some patients and families being more fixated on the device.

## Conclusion

CPI technology was positively perceived by healthcare providers. Patients and their families in general felt the information provided when the monitors were on was being used by nursing staff. These healthcare providers also felt it was easy to use, made them more aware of the need for repositioning, and helped them to target specific areas that needed pressure relief. Most patients and their families did not have any concerns with the device; half were uncertain if the device influenced nursing care. Patient and staff perceptions are not a barrier to future CPI implementation.

## Supporting information

**S1 File. Healthcare provider perceptions survey.**
(PDF)

**S2 File. Patient/family perceptions survey.**
(PDF)

## Acknowledgments

We would like to acknowledge Surakshya Pokharel for performing the literature search and both Nancy Clayden and Cassidy Codan for collecting the study data.

## Author Contributions

**Conceptualization:** Barry W. Baylis, John M. Conly, David B. Hogan, William A. Ghali, Chester Ho.

**Data curation:** Wrechelle Ocampo, Darlene Y. Sola, Linet Kiplagat.

**Formal analysis:** Chester Ho.

**Funding acquisition:** Jaime Kaufman, Chester Ho.

**Investigation:** Barry W. Baylis, John M. Conly, David B. Hogan, William A. Ghali, Chester Ho.

**Methodology:** Chester Ho.

**Project administration:** Wrechelle Ocampo, Jaime Kaufman, Chester Ho.

**Software:** Wrechelle Ocampo.

**Supervision:** Barry W. Baylis, John M. Conly, David B. Hogan, Jaime Kaufman, Henry T. Stelfox, William A. Ghali, Chester Ho.

**Validation:** Wrechelle Ocampo, Chester Ho.

**Writing – original draft:** Wrechelle Ocampo.

**Writing – review & editing:** Wrechelle Ocampo, Darlene Y. Sola, Barry W. Baylis, John M. Conly, David B. Hogan, Jaime Kaufman, Linet Kiplagat, Henry T. Stelfox, William A. Ghali, Chester Ho.

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
