## [Decision Letter · Decision Letter 0]

29 Apr 2022

PONE-D-21-31308Healthcare provider and patient perceptions of continuous pressure imaging technology for prevention of pressure injuries: a secondary analysis of patients enrolled in a randomized control trialPLOS ONE

Dear Dr. Ocampo,

Thank you for submitting your manuscript to PLOS ONE. After careful consideration, we feel that it has merit but does not fully meet PLOS ONE’s publication criteria as it currently stands. Therefore, we invite you to submit a revised version of the manuscript that addresses the points raised during the review process.

Please see the comments from one reviewer below. They are generally positive about the quality and importance of the study, but has suggested some revisions to enhance the clarity and reporting.Please note that we have only been able to secure a single reviewer to assess your manuscript. We are issuing a decision on your manuscript at this point to prevent further delays in the evaluation of your manuscript. Please be aware that the editor who handles your revised manuscript might find it necessary to invite additional reviewers to assess this work once the revised manuscript is submitted. However, we will aim to proceed on the basis of this single review if possible. 

We look forward to receiving your revised manuscript.

Kind regards,

Hanna Landenmark

Staff Editor

PLOS ONE

Journal Requirements:

Reviewers' comments:

Reviewer's Responses to Questions

**Comments to the Author**

1. Is the manuscript technically sound, and do the data support the conclusions?

Reviewer #1: Partly

2. Has the statistical analysis been performed appropriately and rigorously? 

Reviewer #1: No

3. Have the authors made all data underlying the findings in their manuscript fully available?

Reviewer #1: Yes

4. Is the manuscript presented in an intelligible fashion and written in standard English?

Reviewer #1: Yes

5. Review Comments to the Author

Reviewer #1: The subject addressed by this paper, namely the perceptions of healthcare providers, patients, and family members in understanding the acceptability of continuous pressure imaging (CPI) technology, is important and still relevant today. Here are some comments to improve the paper.

Major comments

1. In the section Sampling and subject recruitment authors wrote: “The control group had the ForeSite PT™ system turned on for collecting continuous interface pressure data but the LCD monitor was covered so as not to provide visual feedback to the healthcare providers” What precautions have been taken so that caregivers do not seek to watch the information for the good of their patients?

In this section, authors should indicate and justify their sample size.

2. In the section of Content of Surveys authors should indicate the level of measure of their questions and then indicated if yes or not their data collection tools was validated one.

3. Authors wrote « Closed-ended questions were analyzed using inductive thematic analysis with descriptive statistics using Microsoft Excel » If one can understand that opened-ended questions will be analyse using inductive thematic, however, it seems difficult to apply this to closed-ended questions administrated to more than 125 participants and specially with question on a Likert scale. So, it will be better for authors to describe separately how the analysed qualitative data and quantitative one.

4. Authors wrote “The intervention group had the ForeSite PT™ system set up with the monitor displayed for healthcare workers to view. The control group had the ForeSite PT™ system turned on for collecting continuous interface pressure data but the LCD monitor was covered so as not to provide visual feedback to the healthcare providers. In this case, the healthcare worker provided standard of care to the patient without visual feedback on pressure recordings” This is good, but in following sections such us Analysis section or Results section authors did not refer to intervention group and control group. If they used case control study design they should rewrite analysis section and show how they compare this two groups and, in their results sections, they should used adequate statistics to analysed quantitative data.

If they used another design, so, they should include a study design section and them describe it and adjusted they other section to it.

5. Beware of the desirability bias of patients and healthcare professionals. How was this handled?

6. PLOS authors have the option to publish the peer review history of their article (what does this mean?). If published, this will include your full peer review and any attached files.

Reviewer #1: No

---

## [Author Response · Author response to Decision Letter 0]

23 Jun 2022

1. In the section Sampling and subject recruitment authors wrote: “The control group had the ForeSite PT™ system turned on for collecting continuous interface pressure data but the LCD monitor was covered so as not to provide visual feedback to the healthcare providers” What precautions have been taken so that caregivers do not seek to watch the information for the good of their patients? In this section, authors should indicate and justify their sample size.

Response: In addition to covering the monitors the settings of the monitor were adjusted to a low sensitivity such that the visual feedback from the monitor even if turned on would not be useful. 

• This was added to rows 96-99 on page 6 (“The control group had the ForeSite PT™ system turned on for collecting continuous interface pressure data but the LCD monitor was covered, and the settings were adjusted to low sensitivity, such that the visual feedback to the healthcare providers would not highlight areas of the body in need of pressure relief”). 

How the sample size was determined has now been clarified.

• This information is now found in rows 88-90 on page 6 (“For the RCT, the calculated required sample size was based on the assumption of a 33% relative risk reduction in pressure injuries, an alpha level of 0.05 and 80% power. This lead to an anticipated need for 678 randomized patients with an expected attrition proportion of 12%”).

2. In the section of Content of Surveys authors should indicate the level of measure of their questions and then indicated if yes or not their data collection tools was validated one.

Response: The surveys were developed, reviewed and trialed internally by the study team. 

• This information has been added in rows 116-118 on page 7. Also, copies of the surveys were added as supporting files [“There was no formal validation process for the surveys. Both surveys were developed, reviewed, and piloted within the study team. Copies of both surveys are provided in Supporting Information (S1 File and S2 File)]”.

3. Authors wrote « Closed-ended questions were analyzed using inductive thematic analysis with descriptive statistics using Microsoft Excel » If one can understand that opened-ended questions will be analyse using inductive thematic, however, it seems difficult to apply this to closed-ended questions administrated to more than 125 participants and specially with question on a Likert scale. So, it will be better for authors to describe separately how the analysed qualitative data and quantitative one.

Response: Thank you. We have tried to clarify this point in the revised submission. We clarify to that multiple choice questions were analyzed primarily by descriptive statistics while open-ended questions were analyzed using inductive thematic analysis. 

• The changes can be found in rows 125-129 on page 7 (“Multiple choice questions were analyzed using descriptive statistics in Microsoft Excel, Version 2020. Where deemed appropriate chi-square testing was done. Open-ended questions and free text responses from closed-ended questions were analyzed using inductive thematic analysis (14). These responses were coded manually in Microsoft Excel, Version 2020 and then grouped into themes based on similarities of responses”). 

4. Authors wrote “The intervention group had the ForeSite PT™ system set up with the monitor displayed for healthcare workers to view. The control group had the ForeSite PT™ system turned on for collecting continuous interface pressure data but the LCD monitor was covered so as not to provide visual feedback to the healthcare providers. In this case, the healthcare worker provided standard of care to the patient without visual feedback on pressure recordings” This is good, but in following sections such us Analysis section or Results section authors did not refer to intervention group and control group. If they used case control study design they should rewrite analysis section and show how they compare this two groups and, in their results sections, they should used adequate statistics to analysed quantitative data. If they used another design, so, they should include a study design section and then describe it and adjusted they other section to it.

Response: We performed further analysis (including chi-square testing) and added to our discussion of this point. We show the differences in the distribution of patient/family responses between the control and intervention groups to the question regarding how the system influenced the pressure relief provided by their nurse. These revisions are in:

• rows 38-39 on page 3 (“This response was statistically associated with whether the monitor was turned on (intervention arm; 52.7%) or off (control arm; 4.2%)”); 

• rows 41-42 on page 3 (“Most patients and their families felt it influenced care when the CPI monitor was turned on”); 

• rows 177-180 on page 13 [“Table 4 compared the Control and Intervention group responses to the statement, “the pressure relief provided by my nurse was influenced by the system”. When the monitor was on most respondents (137/260, 52.7%) agreed/ strongly agreed that the system influenced pressure injury care compared to less than one in twenty (11/261, 4.2%) when the monitor was off (chi-square statistic 157, p < 0.001)”]; 

• Table 4 on page 13; 

• rows 208-211 on page 14 (“When the monitor was turned on, most patients and families agreed or strongly agreed that it influenced pressure injury care, while very few felt it did, when the monitor was turned off”); and 

• rows 225-226 on page 15 (“Patients and their families in general felt the information provided when the monitors were on was being used by nursing staff”).

5. Beware of the desirability bias of patients and healthcare professionals. How was this handled?

Response: We believe the anonymity of the healthcare respondents limited desirability bias of in this group.

• Please see rows 219-220 on page 15 (“Desirability bias was limited for the healthcare provider surveys by maintaining anonymity”). 

Desirability bias of patients is indicated as a limitation with the justification that obtaining the data as an in-person interview, as opposed to a self-completed survey, allowed for a better response rate. 

• This is discussed in rows 216-219 on page 15 (“Another limitation is the possibility of desirability bias since the study team was asking the participants the survey questions; however, asking the questions, as opposed to the participant self-completing the survey, allowed for a high response rate and simple yes/no answer questions were avoided”).

---

## [Decision Letter · Decision Letter 1]

12 Aug 2022

PONE-D-21-31308R1Healthcare provider and patient/family perceptions of continuous pressure imaging technology for prevention of pressure injuries: a secondary analysis of patients enrolled in a randomized control trialPLOS ONE

Dear Dr. Ocampo,

Thank you for submitting your manuscript to PLOS ONE. After careful consideration, we feel that it has merit but does not fully meet PLOS ONE’s publication criteria as it currently stands. Therefore, we invite you to submit a revised version of the manuscript that addresses the points raised during the review process.

We look forward to receiving your revised manuscript.

Kind regards,

Yih-Kuen Jan, PhD

Academic Editor

PLOS ONE

Journal Requirements:

Reviewers' comments:

Reviewer's Responses to Questions

**Comments to the Author**

1. If the authors have adequately addressed your comments raised in a previous round of review and you feel that this manuscript is now acceptable for publication, you may indicate that here to bypass the “Comments to the Author” section, enter your conflict of interest statement in the “Confidential to Editor” section, and submit your "Accept" recommendation.

Reviewer #1: All comments have been addressed

Reviewer #2: All comments have been addressed

2. Is the manuscript technically sound, and do the data support the conclusions?

Reviewer #1: Yes

Reviewer #2: Yes

3. Has the statistical analysis been performed appropriately and rigorously? 

Reviewer #1: Yes

Reviewer #2: Yes

4. Have the authors made all data underlying the findings in their manuscript fully available?

Reviewer #1: No

Reviewer #2: Yes

5. Is the manuscript presented in an intelligible fashion and written in standard English?

Reviewer #1: Yes

Reviewer #2: Yes

6. Review Comments to the Author

Reviewer #1: The authors have responded to all comments appropriately. However, a new comment requires a response before the paper is published.

In the Sampling and Subject Recruitment, authors wrote: “For the RCT, the calculated required sample size was based on the assumption of a 33% relative risk reduction in pressure injuries, an alpha level of 0.05 and 80% power. This led to an anticipated need for 678 randomized patients with an expected attrition proportion of 12% (12)”

And in the results section, under Patient/Family Survey sub-section, they wrote: “A total of 525 respondents completed this survey. There were 154 patients/family members who were unable to complete the survey either due to the death of the patient or being discharged from the unit before the survey could be completed.”

So as the sample size is less than the expected one, authors should discuss the implications of that on their results obtained.

Reviewer #2: General comments:

The study aimed to report on the opinions of healthcare providers and patients or family members about CPI.

The study is good to survey the patients and healthcare providers in pressure sensing devices. But, is there any intellectual property (ForeSite PT) to the study?

The researchers may figure out how participants were collected, eliminated and included in the RCT study.

Specific comments:

Line 72: “The main study was a RCT that assessed the effect of the ForeSite PT™ system”

## Might researchers mention the instrument completely (company, state, country)? ## The “a RCT” should be changed into “an RCT.”

Line 73-74: “The system consisted of two parts: a flexible, thin mattress cover with embedded sensors that was positioned under a fitted hospital sheet”

## How do researchers know that the devices have been calibrated (in good condition)?

Line 96-99: “The control group had the ForeSite PT™ system turned on for collecting continuous interface pressure data but the LCD monitor was covered, and the settings were adjusted to low sensitivity, such that the visual feedback to the healthcare providers even, if the monitor was turned on, would not highlight areas of the body in need of pressure relief”

## In this case, how the researchers did “benefit and harm” principles in bioethics?

Line 106: “Content of Surveys”.

## How did the researchers validity and reliability of the questionnaire?

Line 220-221: “Nearly a fifth of healthcare providers had used the CPI technology before. This prior use of the device may have influenced their responses on the use of the device in this study.”

## Why have researchers not excluded the participants (nurses) that have been familiar with the devices before?

For bedsore monitoring to prevent pressure injuries, the authors may consider citing the article below.

## Analysis of Moisture and Sebum of the Skin for Monitoring Wound Healing in Older Nursing Home Residents. In: Vol. 1215 AISC. Advances in Intelligent Systems and Computing (pp. 177-182), 2020

## A time motion study of manual versus artificial intelligence methods for wound assessmen, Plos one, 2022.

7. PLOS authors have the option to publish the peer review history of their article (what does this mean?). If published, this will include your full peer review and any attached files.

Reviewer #1: No

Reviewer #2: **Yes: **Chi-Wen Lung

---

## [Author Response · Author response to Decision Letter 1]

1 Nov 2022

Dear Dr. Jan,

The co-authors and I have gone through the comments and provided revisions. Please see our letter of response and the revised manuscript. Thanks to you and the reviewers for their feedback to improve our manuscript.

Kind regards,

Wrechelle Ocampo

---

## [Decision Letter · Decision Letter 2]

9 Nov 2022

Healthcare provider and patient/family perceptions of continuous pressure imaging technology for prevention of pressure injuries: a secondary analysis of patients enrolled in a randomized control trial

PONE-D-21-31308R2

Dear Dr. Ocampo,

We’re pleased to inform you that your manuscript has been judged scientifically suitable for publication and will be formally accepted for publication once it meets all outstanding technical requirements.

Kind regards,

Yih-Kuen Jan, PhD

Academic Editor

PLOS ONE

Additional Editor Comments (optional):

Reviewers' comments:

Reviewer's Responses to Questions

**Comments to the Author**

1. If the authors have adequately addressed your comments raised in a previous round of review and you feel that this manuscript is now acceptable for publication, you may indicate that here to bypass the “Comments to the Author” section, enter your conflict of interest statement in the “Confidential to Editor” section, and submit your "Accept" recommendation.

Reviewer #1: All comments have been addressed

Reviewer #2: All comments have been addressed

2. Is the manuscript technically sound, and do the data support the conclusions?

Reviewer #1: (No Response)

Reviewer #2: Yes

3. Has the statistical analysis been performed appropriately and rigorously? 

Reviewer #1: (No Response)

Reviewer #2: Yes

4. Have the authors made all data underlying the findings in their manuscript fully available?

Reviewer #1: (No Response)

Reviewer #2: Yes

5. Is the manuscript presented in an intelligible fashion and written in standard English?

Reviewer #1: (No Response)

Reviewer #2: Yes

6. Review Comments to the Author

Reviewer #1: (No Response)

Reviewer #2: The authors have provided a nicely detailed and thorough response to the comments from the previous review and have addressed my major concerns. The authors may consider including more in the discussion section by comparing your results with other related papers.

7. PLOS authors have the option to publish the peer review history of their article (what does this mean?). If published, this will include your full peer review and any attached files.

Reviewer #1: No

Reviewer #2: **Yes: **Chi-Wen Lung

---

## [Editor Report · Acceptance letter]

14 Nov 2022

PONE-D-21-31308R2 

Healthcare provider and patient/family perceptions of continuous pressure imaging technology for prevention of pressure injuries: a secondary analysis of patients enrolled in a randomized control trial 

Dear Dr. Ocampo:

I'm pleased to inform you that your manuscript has been deemed suitable for publication in PLOS ONE. Congratulations! Your manuscript is now with our production department. 

Kind regards, 

on behalf of

Dr. Yih-Kuen Jan 

Academic Editor

PLOS ONE